# Detection and Monitoring of Riverine Dragonfly of Community Interest (Insecta: Odonata): Proposal for a Standardised Protocol Based on Exuviae Collection

**Loan Arguel** [1], **Alice S. Denis** [1,2], **Samuel Danflous** [2], **Nicolas Gouix** [2], **Frédéric Santoul** [1], **Laëtitia Buisson** [1] **and Laurent Pelozuelo** [1,*]

1 Laboratoire d'Ecologie Fonctionnelle et Environnement, CNRS—INPT-UPS, Université Paul Sabatier, Bâtiment 4R1, 118 Route de Narbonne, CEDEX 09, 31062 Toulouse, France

2 Conservatoire d'Espaces Naturels d'Occitanie, BP 57611, 75 Voie du TOEC, CEDEX 3, 31076 Toulouse, France

* Correspondence: laurent.pelozuelo@univ-tlse3.fr

**Abstract:** Collecting quantitative data on insect species occurrence and abundance is a major concern to document population trends. This is especially the case to assess the conservation status of species listed in the European Habitats Directive and to assess the efficiency of mitigation measures with a view to achieve the "no net loss of biodiversity" goal for protected species. However, at present, populations of riverine dragonflies listed in the Habitats Directive and protected under French national law are poorly quantified and monitored. Exuviae collection could be used for such monitoring but a standardised protocol is lacking. We here proposed and tested such a protocol to monitor riverine dragonfly populations through exhaustive exuviae collection along river bank transects. To define the optimal transect size and number of visits, ninety-eight 100 m-long transects divided into 10 m-long plots were monitored on three rivers in southern France. Each transect was visited three times over the emergence period. In the course of each visit, all the exuviae along transects were collected and identified. From our results, we recommend collecting exuviae along 100 m of river bank in the course of two visits in order to both maximise the species detection and minimise the monitoring cost.

**Keywords:** riverine community; survey method; sampling issues; field protocol; conservation; splendid cruiser; pronged clubtail; orange-spotted emerald

## 1. Introduction

Among the ongoing challenges in conservation entomology stands the necessity to acquire quantitative data to document the trends of insect populations [1,2]. Based on long-term quantitative studies, alerts on massive decline of insect populations have multiplied in recent years [3–7]. This situation clearly urges the need for quantitative datasets allowing (i) to detect and monitor local populations, particularly those of threatened or protected species and (ii) to monitor national or supra-national population trends. However, for several dragonfly species, protocols suitable for the production of such quantitative data are simply lacking. This is the case in south-western Europe of the riverine community composed of the Splendid Cruiser *Macromia splendens* (Pictet, 1843), the Pronged Clubtail *Gomphus graslinii* (Rambur, 1842) and the Orange-spotted Emerald *Oxygastra curtisii* (Dale, 1834) [8]. These three species have an unfavourable conservation status at the European community level (Table 1), they are listed in annexes II and IV of the Habitat Directive of the European community, and are strictly protected in Spain, Portugal and France. Moreover, they are southwestern Europe endemic species, their area of distribution being almost limited to France and the Iberian Peninsula [9–12]. This imposes to assess the conservation status of populations at local scale within the Natura 2000 sites and at the national scale [13] (art. 17). It also imposes to offset negative impacts in the case of habitat

alteration or destruction. However, in the context of high anthropogenic pressure on rivers (i.e., increasing demand for infrastructure allowing flood-control, providing irrigation water or producing hydro-electricity), quantitative assessment of both national population trends and local population response to habitat alteration (or restoration) is not possible due to a lack of standardised data. Those species are thus often neglected in the Natura 2000 sites where they are present [14] and the French action plans in favour of dragonflies call for the development of effective standardised protocols [12,15]. To date, national and local environmental authorities, biodiversity consultants in charge of impact assessment studies and nature conservation organisations mostly rely on presence/absence datasets obtained through the compilation of opportunistic sightings. Here, we aimed at providing a standardised protocol to detect those protected species and provide an indicator of their population size.

**Table 1.** The Splendid Cruiser *Macromia splendens*, Pronged Clubtail *Gomphus graslinii,* and Orange-spotted Emerald *Oxygastra curtisii* conservation status and legal status.

| Species | Conservation Status | | French Region [3] | Legal Status | |
|---|---|---|---|---|---|
| | Europe [1] | France [2] | | Europe [4] | France [5] |
| *M. splendens* | VU | VU | VU (Occitanie) EN (Aquitaine) VU (Rhônes-Alpes) | Annexes II et IV | Art. 2 |
| *G. graslinii* | NT | LC | NT (Occitanie) LC (Aquitaine) VU (Rhônes-Alpes) | Annexes II et IV | Art. 2 |
| *O. curtisii* | NT | LC | LC (Occitanie) LC (Aquitaine) LC (Rhônes-Alpes) | Annexes II et IV | Art. 2 |

[1] Kalkman et al., 2010 [16]; [2] UICN France et al., 2016 [17]; [3] Charlot et al., 2018; Barneix et al., 2016; Deliry et al., 2014 [18–20]; [4] European Union Council Directive 92/43/EEC 1992 [13]; [5] Arrêté du 23 April2007 fixant les Listes des Insectes Protégés sur l'ensemble du Territoire et les Modalités de leur Protection; 2007 [21].

Population size can hardly be assessed through adult observation because these dragonflies mature away from emergence sites, males exhibit exacerbated territoriality (especially *O. curtisii*), and imagoes are highly dependent on meteorological conditions for their flying activity. Exuviae collection was investigated as a mean to monitor species of conservation interest, specific richness and odonata community composition since the 2000′s. Foster and Soluk [22] first proved the usefulness of exuviae collection to monitor the population of the endangered Hine's emerald dragonfly, *Somatochlora hineana* Williamson 1931, and Oertli [23] recommended to "prioritize exuviae collection, then larvae and only lastly the adults" to sample Odonata. Hardersen and collaborators in particular compared exuviae collection to larvae collection and adult survey in lotic and lentic habitats [24–26]. Raebel et al. [27] and da Silva-Méndez et al. [28], they showed the different sampling methods are not interchangeable: each has its own advantages and drawbacks. Contrary to adult survey, exuviae provide an unequivocal proof of autochthony and habitat suitability, their collection is poorly or not invasive, and they can easily be used in standardised methods to produce quantitative indicators. Whether or not exuviae collection reduce statistical bias is still controversial [27–29] but it is gaining popularity. There are first attempts to monitor populations of *M. splendens* in Catalunya [11,30,31] through exuviae collection and recently, da Silva-Méndez et al. [28] investigated exuviae persistence time for several riverine species, including *M. splendens*, *G. graslinii* and *O. curtisii.* They confirmed including exuvia collection is essential to assess riverine communities in north-western Iberian peninsula. We thus took advantage of the increased knowledge on exuviae identification—user-friendly identification keys now available, such as Doucet [32] and Boudot & Grand [33]—and increased interest for exuviae collection as a detection and monitoring tool [22,27,34–40] to propose a monitoring protocol based on exuviae collection.

We tested this protocol on three rivers in south-western France (Tarn, Lot and Dourdou de Camarès), during which the sampling distance and the number of visits per season were calibrated. We also provide information about the cost of such a protocol.

## 2. Materials and Methods

### 2.1. Monitoring Sites and Transects Location

Forty-nines sites were sampled on three rivers in south-western France, in the Occitanie region: twenty-one on the river Lot, twenty-six on the river Tarn, and one on the river Dourdou de Camarès (Figures 1 and 2, Table 1 and Table S1). Presence of reproductive populations of *M. splendens*, *G. graslinii*, and *O. curtisii* in at least some stretches of those three rivers was known prior to sampling [9,40,41].

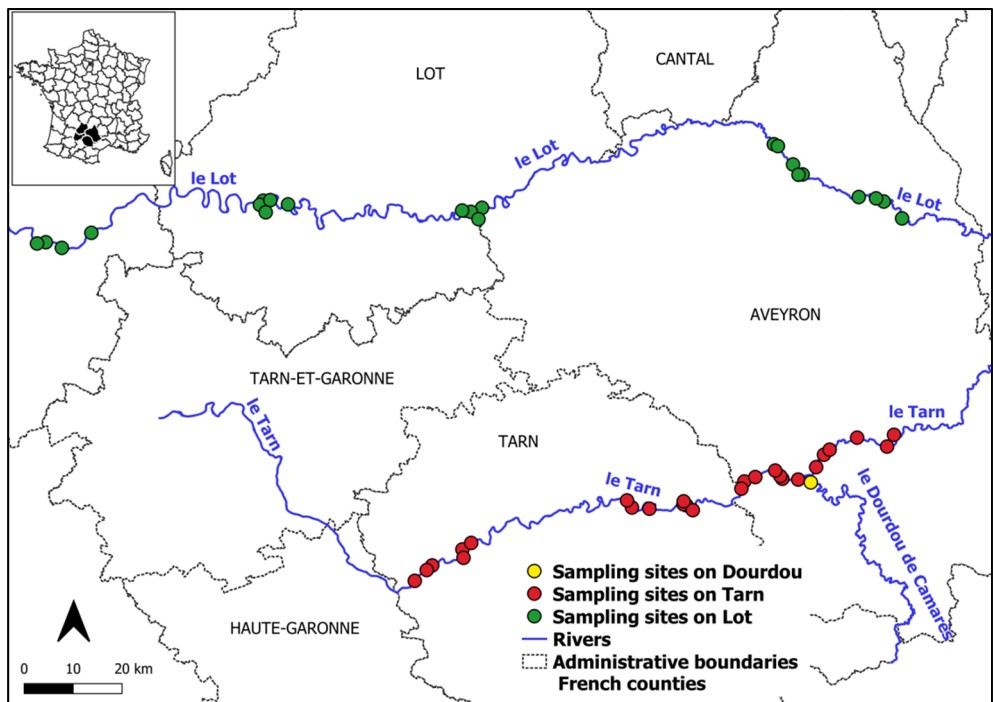

**Figure 1.** Location of study sites in south-western France. Blue lines and blue names indicate the rivers Tarn, Lot and Dourdou de Camarès. Red, green and yellow circles show sampling sites on Tarn, Lot, and Dourdou de Camarès, respectively. The names in capital letters correspond to the French county and their limits are shown in black dotted lines.

Monitoring sites were chosen for their easy access to water. Then, on each monitoring site, two 100 m long river bank stretches (there after named transects) were positioned, one on the left river bank, one on the right river bank. Transects were chosen in a way to fit, as far as possible, the description of the favourable habitats for the targeted species: deep waters, low stream velocity, presence of a dense riparian vegetation with shaded places, or rocky river bank [41–43]. Additionally, we avoided obstacles preventing access to the river bank, such as fallen trees, mud, and sand-banks. A global positioning system handheld (Garmin GPSMAP 65s model) was used to geolocalise the position of each transect and a 20 m long rope with marks every 10 m was used to measure and divide each of them into ten 10 m long plots. For the time of the study, pink warning tape was used to show the beginning and the end of each 100 m transect and each 10 m plot.

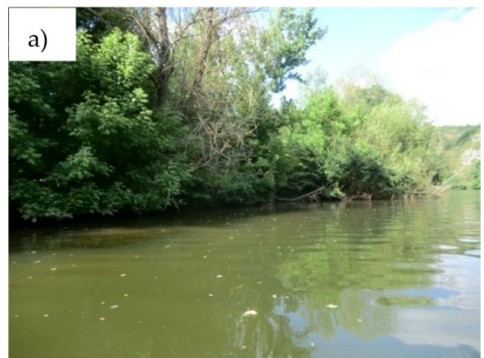
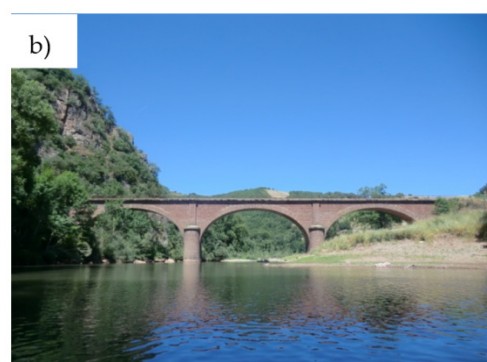
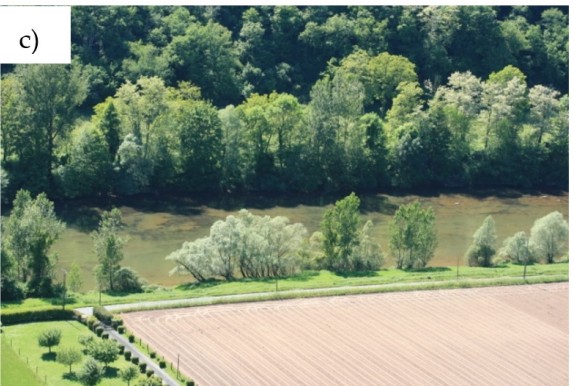

**Figure 2.** Overview of the three rivers sampled: (**a**) Dourdou de Camarès, (**b**) Lot, and (**c**) Tarn.

*2.2. Exuviae Collection*

Along each transect, all Anisoptera exuviae within reach were collected from each 10 m long plot and stored for later identification. Exuviae were collected from a short and easily maneuverable kayak (Mojito model by Rotomod, 250 cm long, 76 cm width and 16 kg weight). The operator visually inspected the riverbank's substrate, herbaceous vegetation and trees roots, trunks, and branches. As emerging larvae can walk high to complete emergence (Arguel, Pelozuelo and Denis, personal observation), trunks, branches and mineral cliffs and rocks were scouted from their base to approximately 3 m high. Exuviae found at such height were collected using the 2.2 m long kayak paddle. On each transect, the total sampling time and at each 10 m portion was recorded.

Each site was sampled three times from June to August 2015 to cover the entire expected emergence period of the riverine community in south-western France (Table S1). The first visit to each transect took place between 9 June and 14 July, the second visit between 9 July and 7 August, and the third one between 27 July and 26 August (Table S1). The date for the first visit were chosen to be around the peak of emergence, based on information's available at that time on Vère and Aveyron rivers (Denis and Pelozuelo, personal observation) and later confirmed [44]. However, it is noteworthy that local phenology is not described on those rivers, and emergence can even occur later due to deep and thus cold waters of the Lot and Tarn rivers, as observed on Tarn river in 2020 and 2021 [45]. The delay between each visit on the same site was on average, 25 days between the first and the second visit (Min = 15, Max = 42, Median = 23) and 18 days between the second and the third visits (Min = 11, Max = 23, Median = 20). We could not have shorter delays between the visits due to rainy weeks.

Given that *M. splendens*, *G. graslinii*, and *O. curtisii* are protected species in France, the exuviae collection was authorised by prefectural decrees n°81-2014-05 and n°82-2014-05. Collected exuviae were identified in the laboratory, using a binocular microscope (Leica Zoom 220 model) and the identification key of dragonfly exuviae of France [32]. Numbers of exuviae per species, per 10 m plot of each transect were then obtained.

*2.3. Practical Considerations: Time and Cost Required for Such a Protocol*

The time required to sample each 100 m long transect was measured, from the beginning to the end of each transect. The cost of such a study has also been estimated based on the prices of the different material items required, as given on several websites. The costs of the equipment specifically required to this study (navigation equipment, roof bars, etc.) were estimated separately from the cost of the basic equipment generally present in an ecology laboratory (binocular microscope, car, identification key, etc.).

*2.4. Data Analysis*

For each 10 m plot, 100 m transect, or site (left plus right river bank transects), species richness and abundance were calculated. We then calibrated the sampling effort (i.e., number of visits and length of transects) that would be required to maximise abundance, species richness, and detection of the target species (i.e., *M. splendens*, *O. curtisii*, and *G. graslinii*). We calculated the number of new species detected and the abundance of exuviae collected at each visit to assess the efficient number of visits. We compared them using a Friedman rank sum test, followed by a non-parametric Wilcoxon signed-rank test with Bonferroni adjustment to find post-hoc statistical differences. We also calculated the cumulated richness detected by plots. Cumulated species richness was plotted to calibrate the effective length of transects. We finally tested the correlation between exuviae abundance by transect and sampling duration using the Spearman correlation test. All analyses were performed with R 4.0.3 (R Core Team, 2020) and all maps were made with QGIS Desktop 3.12.3.

Species that were poorly present in our samples of exuviae (i.e., less than 10 exuviae across all exuviae collected over the 98 transects) and exuviae that we could not identify were not included in the analysis. As *M. splendens* was one of the species of community interest we focused on, its exuviae number are shown even if they were below the fixed threshold. For each species, densities of exuviae per 100 m for first plus second visit were calculated.

## 3. Results

In total, 5831 exuviae from 11 Anisoptera species (with more than 10 exuviae each) were collected. Six species are typically riverine according to the description of their habitats [10,33], two are occasionally riverine, and three are rather associated with standing waters. Besides the species targeted by this study (i.e., *M. splendens*, *G. graslinii* and *O. curtisii*), the community found in this region included *Onychogomphus forcipatus* (Linnaeus, 1758), *Boyeria irene* (Boyer de Fonscolombe, 1838), *Gomphus vulgatissimus* (Linnaeus, 1758) as riverine species, plus *G. pulchellus* (Selys, 1840), and *Somatochlora metallica* (Vander Linden, 1825) as occasionally riverine and *Orthetrum cancellatum* (Linnaeus, 1758), *Anax imperator* (Leach, 1815), and *A. parthenope* (Selys, 1839) as rather associated with standing waters. *Aeshna mixta* (Latreille, 1805) (6 exuviae), *Cordulegaster boltonii* (Donovan, 1807) (3 exuviae), *Gomphus similimus* (Selys, 1840) (1 exuvia), *Libellula fulva* (O.F Müller, 1764) (1 exuvia), *Orthetrum albistylum* (Selys, 1848) (1 exuvia), *Orthetrum brunneum* (Boyer de Fonscolombe, 1837) (1 exuvia), *Sympetrum sanguineum* (OF Müller, 1764) (1 exuvia), *Sympetrum striolatum/meridionalis* (Charpentier, 1840 and Selys, 1841) (10 exuviae), and *Trithemis annulata* (Palisot de Beauvois, 1807) (4 exuviae) were also detected but given the low number of exuviae collected from each of these species, they were discarded from further analysis. The rivers sampled are relatively wide and deep (around 90 m for the Lot, 100 m for the Tarn and 30 m for the Dourdou-de-Camarès) and thus the scarcity of *C. boltonii* is not surprising as it prefers smaller and shallower tributaries.

Furthermore, there were some exuviae that could not be identified to the species level and thus were not considered in the analyses: *Anax* sp. (28 exuviae), *Aeshna* sp. (1 exuvia), *Gomphus* sp. (10 exuviae), and *Sympetrum* sp. (3 exuviae).

The presence of the three species of community interest in different places of the studied rivers (Tarn, Lot and Dourdou de Camarès) was confirmed. *O. curtisii* and *G. graslinii*

were found on the Tarn and the Lot, with relatively greater numbers of the downstream part of these two rivers (Tables 2 and 3 and Figure 3). *M. splendens* was detected on two sites on the Tarn and one on the Lot. Density of *M. splendens* was much lower compared to *O. curtisii* and *G. graslinii*. On the Dourdou de Camarès river banks, only *O. curtisii* was detected.

**Table 2.** List of dragonflies species with more than 10 exuviae detected on each river (Lot, Tarn, and Dourdou de Camarès) with the total number of transects where the species was detected at least once and the associated percentage of positive transects (between parentheses). Riverine species according to Dijkstra [10] and Boudot and Grand [33] are indicated by asterisks and this study's target species (i.e., protected ones) are indicated in bold.

| Odonata Species | Rivers | | |
|---|---|---|---|
| | Lot (*n* = 44) | Tarn (*n* = 52) | Dourdou de Camarès (*n* = 2) |
| ***O. curtisii** | 22 (50) | 16 (30.8) | 1 (50) |
| ***M. splendens** | 1 (2.3) | 2 (3.8) | - |
| ***G. graslinii** | 27 (61.4) | 24 (46.2) | - |
| *G. vulgatissimus* | 23 (52.3) | 31 (59.6) | 2 (100) |
| *O. forcipatus* | 34 (72.3) | 45 (86.5) | 2 (100) |
| *B. irene* | 26 (59.1) | 27 (51.9) | 1 (50) |
| S. metallica | 12 (27.3) | 6 (11.5) | - |
| O. cancellatum | 16 (36.4) | 1 (1.9) | - |
| G. pulchellus | 5 (11.4) | 6 (11.5) | - |
| A. imperator | 4 (9.1) | 2 (3.8) | - |
| A. parthenope | 4 (9.1) | 2 (3.8) | - |

**Table 3.** Average exuviae density per 100 m long transects obtained during the first plus second visits for each species detected on each river (Lot, Tarn, and Dourdou de Camarès). The corresponding standard deviations are shown between parentheses. Riverine species according to Dijkstra [10] and Boudot and Grand [33] are indicated by asterisks and this study's target species (i.e., protected ones) are indicated in bold.

| Odonata Species | Rivers | | |
|---|---|---|---|
| | Lot (*n* = 44) | Tarn (*n* = 52) | Dourdou de Camarès (*n* = 2) |
| ***O. curtisii** | 29.4 (91.5) | 11.7 (26.7) | 3 (4.2) |
| ***M. splendens** | 0.02 (0.2) | 0.1 (0.3) | - |
| ***G. graslinii** | 8.8 (13.8) | 10.7 (24.5) | - |
| *G. vulgatissimus* | 6.3 (18.1) | 5.9 (11) | 8.5 (6.4) |
| *O. forcipatus* | 10.6 (14.9) | 17.6 (17.2) | 94.5 (36.1) |
| *B. irene* | 5.2 (13.3) | 2.2 (3.2) | 2.5 (0.7) |
| S. metallica | 0.7 (1.4) | 0.2 (0.6) | - |
| O. cancellatum | 1.8 (5.4) | 0.02 (0.1) | - |
| G. pulchellus | 0.3 (1.1) | 0.2 (0.6) | - |
| A. imperator | 0.05 (0.2) | 0.04 (0.2) | - |
| A. parthenope | 0.02 (0.2) | 0.1 (0.3) | - |

### 3.1. Number of Visits

On the three visits carried out, it appears very clearly that the first visit was the most informative in terms of both specie richness and abundance (Figures 4 and 5. Indeed, the number of new species detected on a site was significantly much lower during the second (Friedman test, $p < 0.001$; Wilcoxon signed-rank test, P1–2 < 0.001) and the third visits (Friedman test, $p < 0.001$; Wilcoxon signed-rank test, P1–3 < 0.001, P2–3 = 0.01; Figure 4).

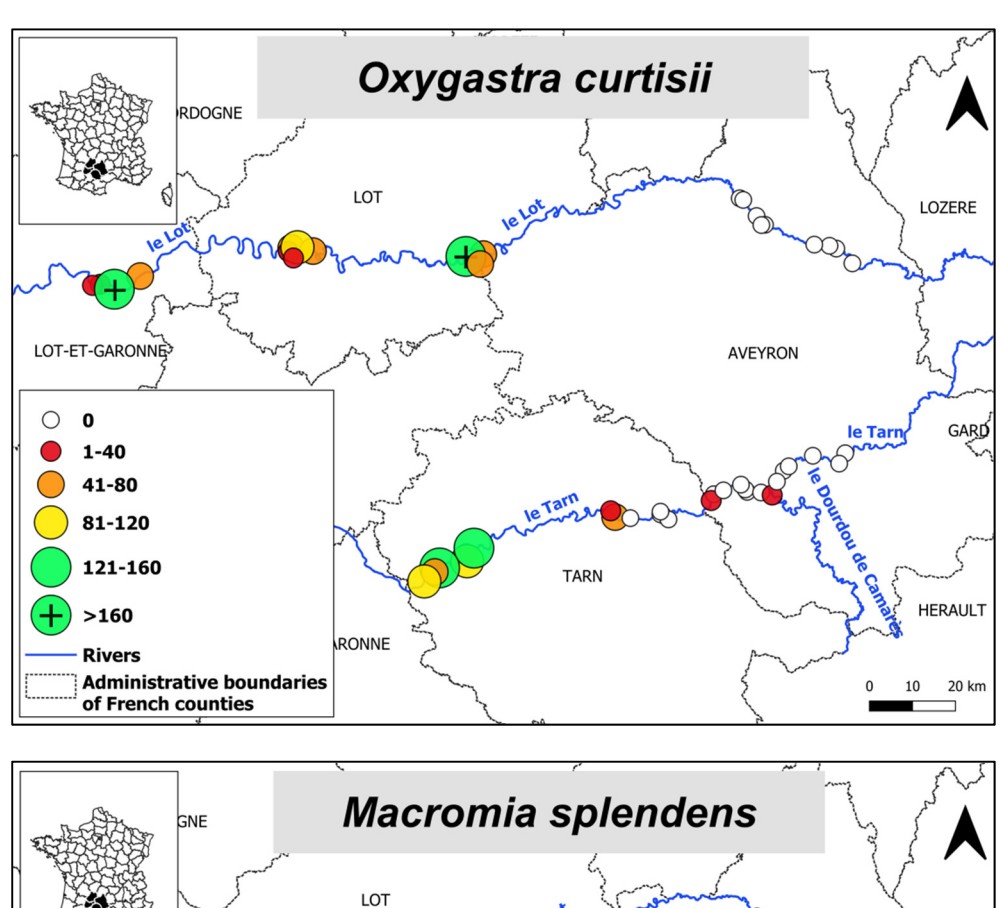

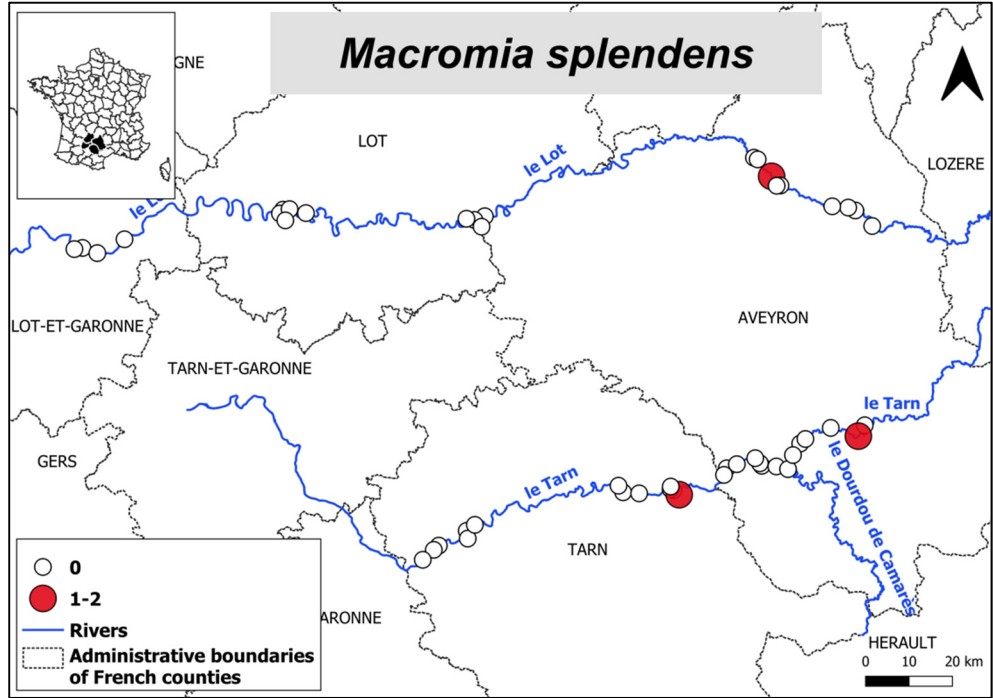

**Figure 3.** *Cont.*

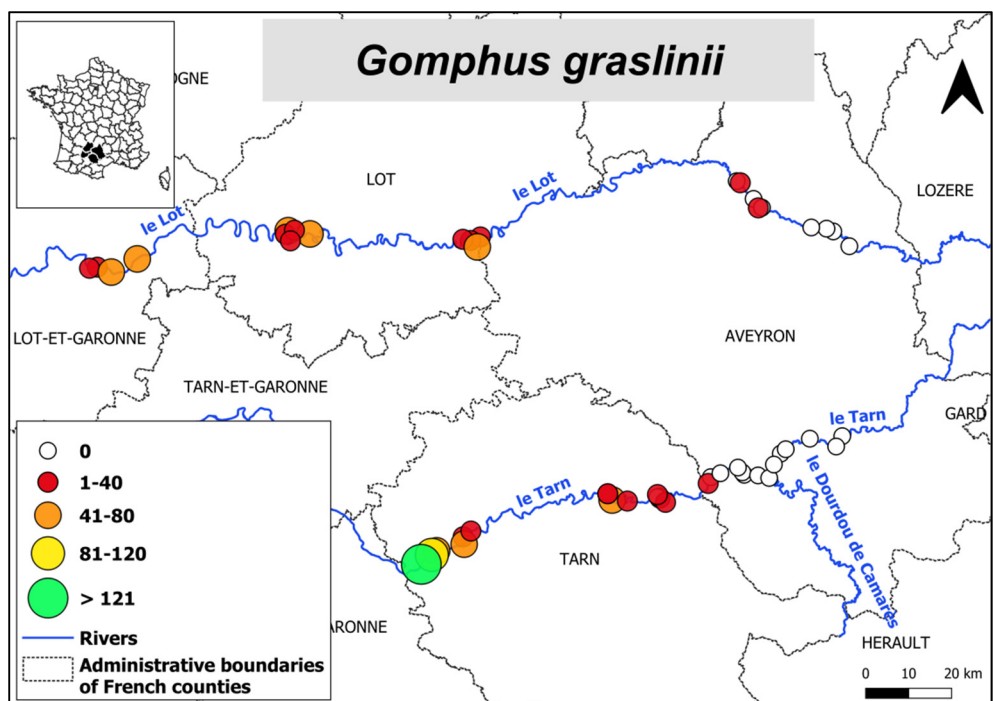

**Figure 3.** Distribution maps of exuviae of the three species of community interest *Oxygastra curtisii*, *Macromia splendens* and *Gomphus graslinii*. Bold lines and bold names indicate the rivers Tarn, Lot, and Dourdou de Camarès. The gradient of colour and size of the circles highlights the differences in numbers of exuviae.

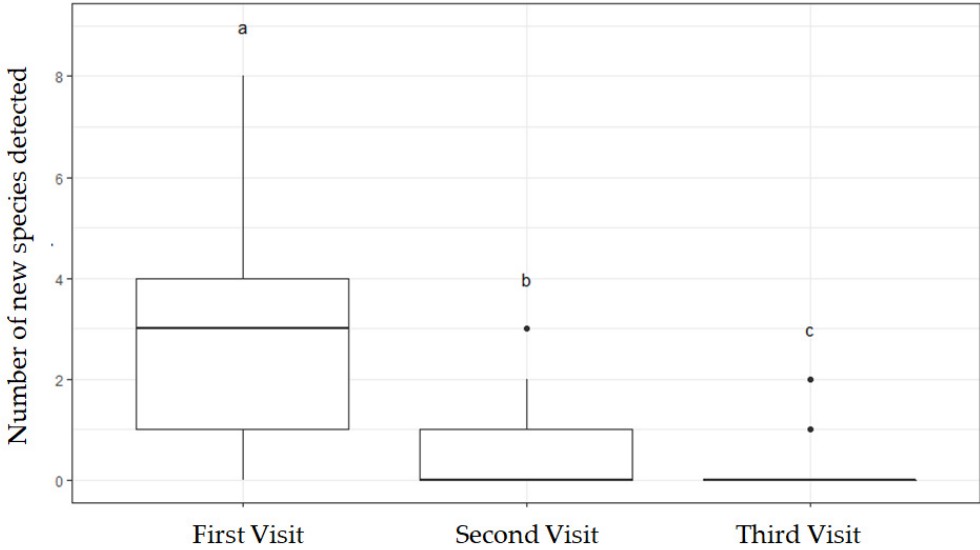

**Figure 4.** Number of new species detected calculated by transect (*n* = 98) depending on the number of visits. Box plots indicate median, range, and first and third quartiles. Points indicate outliers. Significant differences are indicated by different letters.

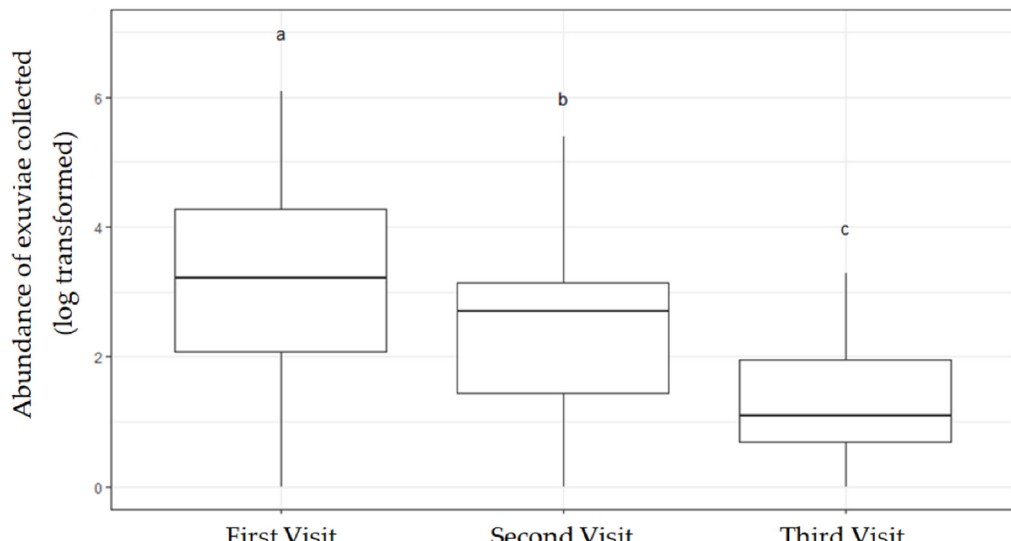

**Figure 5.** Abundance of exuviae collected (all species pooled) per 100 m long transects (*n* = 98 transects) depending on the visit rank. Abundances are log-transformed. Box plots indicate median, range, and first and third quartiles. Significant differences are indicated by different letters.

The species richness detected during the first visit represented on average 82.8% of the total detected richness on a site. This percentage fell respectively to 12.8% and 4.4% during the second and the third visits. Moreover, the total abundance of exuviae was significantly higher during the first visit than during the other two (Friedman test, $p < 0.001$; Wilcoxon signed-rank test, P1–2 < 0.001, P1–3 < 0.001, P2–3 < 0.001; Figure 5).

The three species of community interest were mostly detected during the first and second visits. For *O. curtisii*, detection occurred during the first visit on 94.9% of the transects where the species was detected at least once during the entire sampling period (*n* = 39), for *G. graslinii* on 86.3% of transects (*n* = 51) and for *M. splendens* on 66.7% of transects (*n* = 3) (Figure 6). The second visit allowed detecting *G. graslinii* for the first time in 11.8% of cases, *M. splendens* in 33.3% of cases, and *O. curtisii* in 5.1% of cases. In the third visit, only *G. graslinii* was detected on a single transect (on the Lot river) among the 51 sampled (Figure 6).

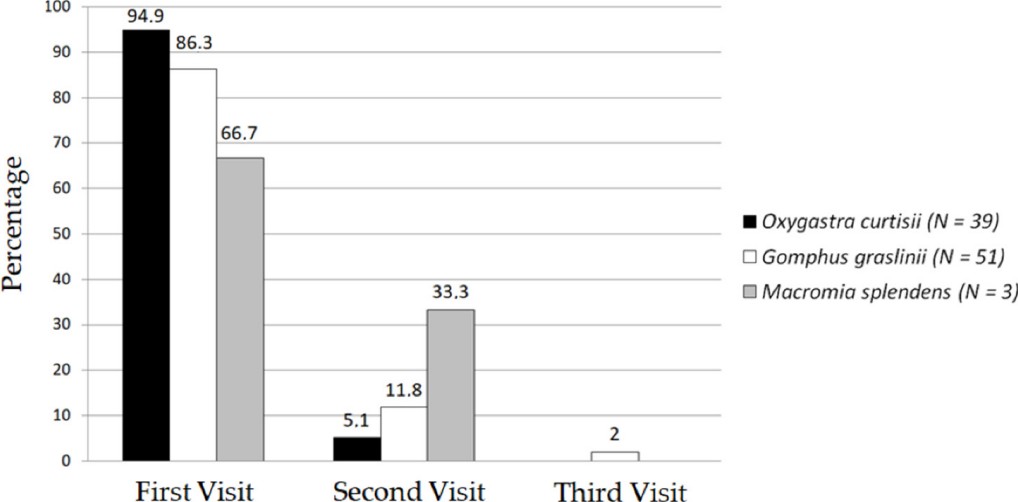

**Figure 6.** Distribution of the first detection for the three species of community interest, *Gomphus graslinii*, *Macromia splendens*, and *Oxygastra curtisii*, across the first, second, and third visits.

The first visit enabled us to collect 69.4% of the total exuviae collected, while the second and the third ones represented respectively 25.5% and 5.1% of the total. With the two first visits, we thus detected about 95.6% of the total species richness and collected 94.9% of the total abundance.

Regarding the three species of community interest, the abundance of exuviae was significantly higher during the first passage and it significantly decreased at the second and third visit (for *G. graslinii*: Friedman test, $p < 0.001$; Wilcoxon signed-rank test, P1–2 < 0.01, P1–3 < 0.001, P2–3 < 0.001; for *O. curtisii*: Friedman test, $p < 0.001$; Wilcoxon signed-rank test, P1–2 < 0.001, P1–3 < 0.001, P2–3 < 0.001) (Figure 7). The number of *M. splendens* exuviae was too small (four exuviae) to carry any analysis.

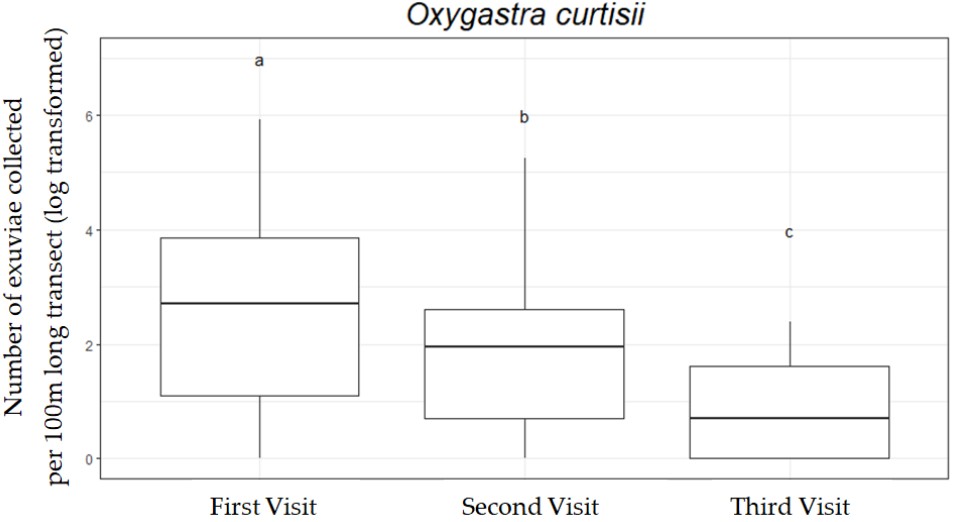

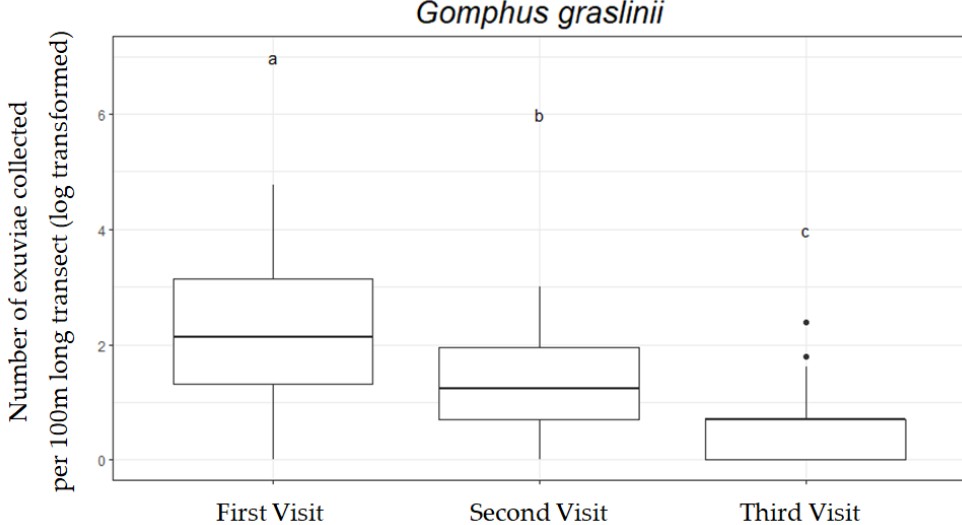

**Figure 7.** Abundance of exuviae of *Gomphus graslinii* and *Oxygastra curtisii* collected along 100 m-long transects (*n* = 98 transects) during the first, second, and third visits. Abundances are log-transformed. Box plots indicate median, range, and first and third quartiles. Points indicate outliers. Significant differences are indicated by different letters.

### 3.2. Transect Length

Along the 100 m transects, the number of new species detected mostly increased within the first 70 m then tended to stabilise. The first four plots allowed, on average, detection of over 70% of the species. The arbitrary threshold of 90% of detected species richness was reached at, on average, between 60 and 70 m (Figure 8).

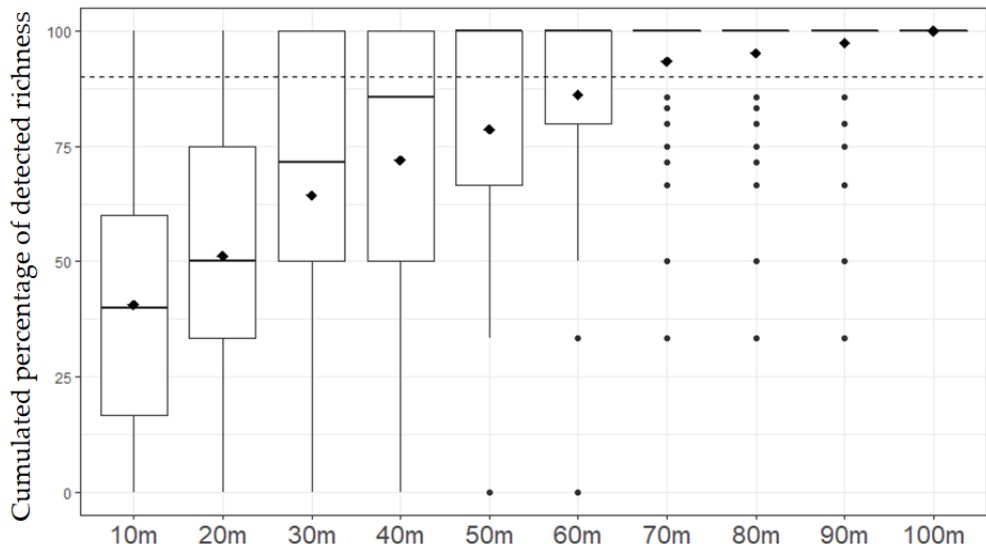

**Figure 8.** Cumulated percentage of detected richness depending on number of 10 m plots scouted for exuviae along each transect (*n* = 98 transects). Box plots indicate median, range, and first and third quartiles. Points indicate outliers. Black diamonds indicate the mean of each section. Dashed line indicates the 90% threshold.

Regarding the three species of community interest, detection generally took place within the first 10 m plots. *G. graslinii* and *O. curtisii*, are detected in the first 10 m plot in 47.1% and 59% of positive transects respectively (Figure 9). Beyond 60 m, the first detection of *G. graslinii* and *O. curtisii* was very low (<8%). The number of *M. splendens* was too low to make any conclusion.

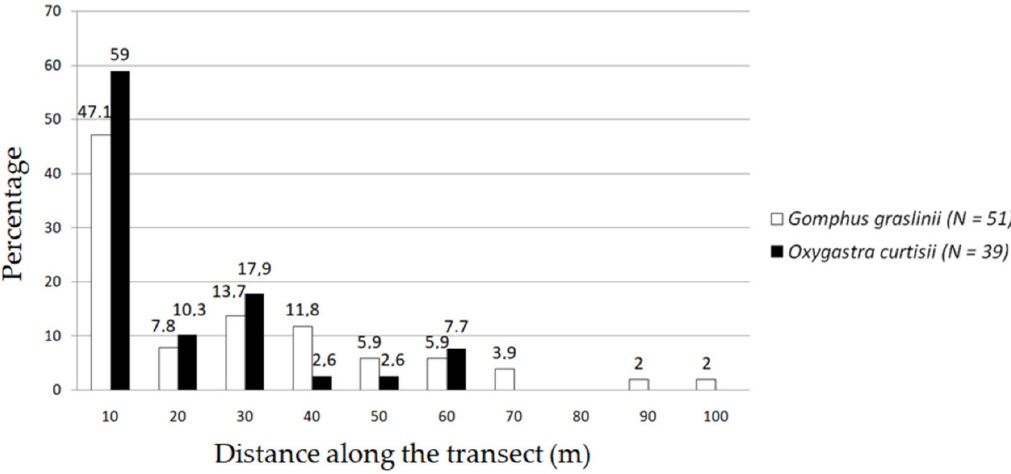

**Figure 9.** Distribution of the first detection of *Gomphus graslinii* and *Oxygastra curtisii* along the 100 m long transects (*n* = number of positive transects for each species).

*3.3. Practical Considerations: How Long Did It Take and How Much Did It Cost?*

On average (*n* = 98), exuviae collection along one 100 m long transect required 1 h 14 min (Max = 2 h 11 min and Min = 29 min) during the first visit, and, respectively, 52 min (Max = 1 h 37 min and Min = 28 min) and 39 min (Max = 1 h 7 min and Min = 14 min) during the second and the third visits (Table S2). The duration of exuviae collection is significantly and positively correlated with exuviae abundance (Spearman's rank correlation rho = 0.78, *p* < 0.001). This is the raw time necessary to collect exuviae but the time to handle, load, and unload the kayak on the car roof must be added (approximately 20 min) as well as the time necessary to drive from the lab to the different sites. Within normal

conditions, two persons on two separate kayaks can sample three sites per day (i.e., six transects per day).

Equipment cost for exuviae collection (one kayak and its accessories such as a paddle, a lifejacket, a 5 L waterproof container, containers for exuviae collection, one handheld GPS, and a 20 m-long rope) is around €1000 per person and two persons are required for security reasons. The material cost for exuviae identification (binocular microscope, identification key, etc.) is around €850.

## 4. Discussion

This study aimed to propose a standardised protocol for monitoring riverine dragonfly communities. We investigated the optimal number of visits to pay to each transect and the optimal length of transect to accurately describe the species composition and abundance of the local Anisoptera community, with a special focus on species of community interest, *O. curtisii*, *M. splendens*, and *G. graslinii*. Regarding the number of visits per transect, our results showed that two visits are required to detect the majority of species and specifically our three species of community interest. Even if most species were detected during the first visit, a second visit brought valuable additional information as *O. curtisii* and *G. graslinii* would have been not detected in 5.1% and 11.8% of our positive transects without a second visit. On the contrary, there is no need for a third visit as the number of new species detected during this visit was nearly zero and, if not, new species detected were species with no conservation issue, e.g., *A. parthenope*, *T. annulata*, or *L. fulva*. Our conclusion is similar for abundance: two visits allowed us to collect around 94.9% of the total number of exuviae. On the contrary, the proportion of exuviae collected during the third visit was low (5.1%). This pattern might be the result of our visit schedule: the first visit probably occurred around the emergence peak, the second visit at the end of the emergence period—allowing collection of the exuviae of the individuals emerging lately—and the third visit may have occurred when the emergence period was already finished, and thus only exuviae unseen during the first and second visit were remaining.

Therefore, we conclude that it is not useful to carry out a third visit, especially since it would occur lately in the season and would probably not allow the detection of species of community interest which are all early species [41,46]. As sampling effort is always a trade-off between the search for exhaustiveness and time and money allocated, we recommend two visits rather than three.

We did not investigate the delay between each visit. Even if we recognise that this would deserve more attention, the 18- up to 25-day-long interval between first and second visit seems appropriate. Such a delay allows to stop sampling and to wait for new emergence in case of unpredictable meteorological event such as a storm or a flood washing away the exuviae. During this study, we had to face periods of heavy rains which obliged us to stop exuviae sampling and increased the delay between the first and second visit. This probably had a negative impact on the exuviae densities [27]; however, this was not quantified in our case and, hence, we cannot provide a solution to manage the impact of such events.

We also investigated the effect of transect length on our ability to detect species. Our results showed that a 70 m-long transect would be enough to detect, on average, 90% of the species locally present on each transect. This particularly applies for *O. curtisii* and *G. graslinii* which were detected within the first 10 m plots in a vast majority of positive transects, 87% and 69% respectively within the first 30 m. Concerning *M. splendens*, we cannot draw any conclusion because this species was too scarce on our transects (only three positive transects, with a total of four exuviae), even though the species had been previously observed in various location of the Tarn and Lot rivers [41,47–49]. We would recommend to sample 100 m-long transects rather than 70 m-long ones. First, sampling 100 m-long transects would only last 15 min more on average. Second, a 100 m-long distance is easy to remember and handle in our decimal metric system. Third, we can expect a 100 m-long transect to increase our chances to detect *M. splendens*.

Compared with the results of da Silva-Méndez in the NW Iberian Peninsula [28] and those of samplings we have carried out in other French rivers (Aveyron, Vère, Viaur, unpublished data), exuviae densities were surprisingly low on some sites, particularly for *M. splendens*, whose exuviae were occasionally collected by hundreds on Tarn river in the 1980′s [41]. Such a situation has poor chance to be due to late collection date as our first visit took place around the date 50% of emergence occurred on close by rivers [44]. Furthermore, in a recent survey [45], around fifty *M. splendens* exuviae could be found on Tarn river between mid-July and early August. As adults damselflies were also few (Denis, Pelozuelo and Danflous, personal observation), we rather think it is due to a low level of Odonata populations at that time. Dam emptying operation at the Pinet dam in 2003 and 2009, with the water level moving down by 9 m and 15 m respectively [50], could have heavily impacted Odonata populations on the upper Tarn river for several years. Heavy rains that occurred at the end of June could also washed away an important part of the exuviae. Anyway, locally low densities of exuviae do not undermine our results and both the transect length and the two visit we recommend would be enough for an accurate description of dragonflies community in rivers with higher densities.

Differences in detection between adults and exuviae of Odonata have been shown in recent studies and the exuviae collection is today one of the most reliable methods for monitoring species [25,29]. Moreover, exuviae detection offers numerous benefits: on one hand, exuviae are "the most important indicators of resident populations" [51], i.e., the best cue of on-site reproduction and development [27,41,52] and on the other hand, the number of exuviae provides the most reliable estimate of population density [23]. In addition, exuviae collection is a non-invasive method, an essential quality when dealing with protected or red-listed species [22]. Indeed, exuviae collection has become in recent years a popular sampling method to inventory Odonata in lentic [22,26] and lotic habitats [26,40,53,54]. However, few standardised protocols based on the exuviae collection have been investigated. There are examples of exhaustive exuviae collection during a defined duration [40], of exhaustive exuviae collection over transects randomly drawn each year [53,55,56] or over chosen "sentinel" sites to be monitored every year ([8], this study). Number of visits and transect length differ from one protocol to another; however, such differences are adaptations to local conditions. In our case, riverbanks are not drastically modified from one year to another in the rivers monitored and our protocol can easily be applied. With a good knowledge of the target species' emergence periods, two visits on identical 100 m-long transects every year are enough to monitor riverine dragonfly communities in a way that allows to obtain trends after several years of monitoring. Thanks to the use of "sentinel" sites, sampled each year, spatial and temporal variations in exuviae densities can be highlighted. As important interannual variation might be expected, several years of monitoring would be recommended to establish a local reference.

However, for this type of protocol, additional recommendations can be made. First, as a feedback from our own field-work experience with kayak for exuviae collection, the use of short individual kayaks would be recommended, even on small rivers, since it allows good access to the river banks regardless of water levels, without trampling the aquatic habitat and larvae. It should however be noted that in some rare cases kayaks may become impractical in small Mediterranean rivers, restricted to pools during part of the summer. Additionally, cleaning the kayak and accessories between sites/sampling dates may be required to prevent the spread of invasive aquatic species if such a risk is identified.

For safety purposes, surveys should always be carried out in pairs. Kayaks also enable sampling to be carried out in a comfortable position (sampling sessions can sometimes last all day) and to access areas that are difficult to access on foot (e.g., steep banks, sections of isolated rivers). Then, it is important to plan sampling according to meteorological and hydrological conditions. Exuviae can be washed by storms (rainfall and gusts of wind) or when water levels rise (because of dam water release or precipitations) [57]. Thus, we recommend leaving a few days after those occasional disturbances, to allow larvae to emerge.

Standardised protocols based on exuviae collection are currently the most reliable and relevant methods for detecting and monitoring riverine dragonflies. They should be implemented to improve knowledge of targeted species population trends and ensure their conservation, as recommended by the latest French National Action Plan for dragonflies [12]. In the future, they should be used to test and develop other methods such as quantitative environmental DNA approaches. Efforts to produce genetic barcodes for species identification [58,59] and to understand the persistence and accumulation of dragonflies DNA in their ecosystems [60] would probably soon make quantitative environmental DNA approaches available. Using standardised exuviae collection with such genetic methods will thus be useful to calibrate and validate the use of genetic methods to monitor odonate populations in nature.

Finally, we should emphasise that this protocol has been used every year since 2018. Except for the impossibility to quantify how much storms may impact exuviae densities, no particular drawback has been identified since, and the dataset obtained is being analysed in order to describe temporal and spatial variability for these three targeted species and identify rivers and rivers portion of highest conservation value.

## 5. Conclusions

The extensive sampling effort set up during this study has allowed us to propose and calibrate a relevant protocol for surveying riverine dragonfly communities. Even though sample sites were only located on three rivers in south-western France, this method may be suited to abroad range of temperate rivers. Thus, we propose riverine dragonfly surveys to be conducted by two observers in kayaks (i.e., one along each bank) in the course of two visits during the emergence period to collect exuviae of all species along a 100 m transect of river bank. According to our results, this method maximises detected richness while minimising the duration of sampling. These results are a major issue since exuviae collection is rarely undertaken when surveying riverine dragonflies, especially as no standardised monitoring program currently exists for the three protected species in France (i.e., *O. curtisii*, *M. splendens*, and *G. graslinii*) and only recently effort were also dedicated to develop a protocol for the monitoring of *Ophiogomphus cecilia* (Geoffroy in Fourcroy, 1785) and *Stylurus flavipes* (Charpentier, 1825) [12,53]. This is also particularly true regarding environmental impact assessments, which aim to avoid, mitigate, and offset adverse impacts on biodiversity, and particularly on protected species. Methods for detecting riverine dragonflies and quantifying their populations should be relevant and robust to ensure that decision-makers' judgments are well-founded. Thus, we expect our protocol proposal to raise awareness among the experts involved in impact assessment and the administration to review their studies.

**Supplementary Materials:** The following supporting information can be downloaded at: https://www.mdpi.com/article/10.3390/d14090728/s1, Table S1: Location and sampling dates of the ninety-eight transects. GPS points are provided in WGS84 geographic format; Table S2: Details of survey durations on the different rivers (average, standard deviation, maximum duration and minimum duration).

**Author Contributions:** Conceptualization, A.S.D., S.D. and L.P.; methodology (field data collection), A.S.D. and L.P.; methodology (analysis data collection), S.D.; formal analysis, A.S.D. and L.B.; figures preparation, L.A.; writing—original draft preparation, A.S.D. and L.P.; editing, L.A.; final draft writing, L.A. and L.P.; supervision, N.G., F.S. and L.P. All authors have read and agreed to the published version of the manuscript.

**Funding:** This research was funded by Agence Nationale de la Recherche et de la Technologie (Cifre PhD grant n° 2015/0051), Electricité de France (EDF), European Regional Development Fund (FEDER), Agence de l'Eau Adour-Garonne and a donation by Mrs. MANN. The APC was funded by Laboratoire d'Ecologie Fonctionnelle et Environnement.

**Institutional Review Board Statement:** Not applicable.

**Data Availability Statement:** Not applicable.

**Acknowledgments:** We thank the following people for their respective contributions to this work: Baptiste Charlot and Laetitia Pêcheur for their great help to identify the exuviae and also Marine Valet and Maxime Sacré for their help in the data collection.

**Conflicts of Interest:** The authors declare no conflict of interest. The funders had no role in the design of the study; in the collection, analyses, or interpretation of data; in the writing of the manuscript, or in the decision to publish the results.

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
