# Peer review of "Detection and Monitoring of Riverine Dragonfly of Community Interest (Insecta: Odonata): Proposal for a Standardised Protocol Based on Exuviae Collection"

_diversity, doi:10.3390/d14090728_

Round 1
Reviewer 1 Report
Title: Detection and monitoring of riverine dragonfly of community interest (Insecta: Odonata): proposal for a standardised protocol based on exuviae collection
Reference: diversity-1855617
The authors present a standardized methodology by transects to monitor species of Anisoptera (Odonata) included in the Habitats Directive (Macromia splendens, Oxygastra curtisii, Gomphus graslinii). It is a useful approach to the problem, although some issues have not been sufficiently addressed in the manuscript. The period of the year in which the first visit to the river is made, the overlap between the first and second visits, or between the second and third visits, in different rivers should be discussed. The lack, or scarcity, of exuviae of some species (M. splendens included) has not been explained in the manuscript.
Comments:
In data analysys section of Methods, authors erase species poorly present in exuviae samples (lines 159-168)... this is suitable, but the low presence of Cordulegaster boltonii in the samples was very strange. It is a common species in rivers, and, in my experience in Portugal and Spain, it is usually present in the sections where the three target species of the work are present (M. splendens, O. curtisii and G. graslinii) (Not being the other way around, since C. boltonii is much more frequent in the rivers of the Iberian Peninsula, including where the species of the Habitats Directive are not found). For example, in the North of Portugal: Torralba-Burrial, A., Silva, G. D., Rodríguez-Martínez, S., Menéndez, D., García García, I., Fernández González, Á., & Fernández Menéndez, D. (2013). Las comunidades de libélulas de la cuenca media-alta del río Támega (NE Portugal)(Insecta: Odonata). Boletín de la Sociedad Entomológica Aragonesa, 52: 173–190. Is there any reason why this is not the case on these French rivers? According to Boudot, J-P, & Kalkman (eds.) 2015 Atlas of the European dragonflies and damselflies. KNNV publishing, the Netherlands is also very frequent in these rivers.
Results
Lines 196-204: There is a problem with the text and figure/table mentions.
line 156-7 Figure 2. Authors write "The size of the circles is proportional
to the total number of exuviae collected on a site (i.e. two transects) during the three visits", I think is the total number ot exuviae of each species collected on a site. I think the scale could be improved. The aim of the map is to show different realities between reachs, when describing the exuviae data. The location of the exuviae of M. splendens has been anecdotal in this work, and there are no practical differences between finding one or two exuviae in the section, so it makes no sense that they are shown differently in the figure.
Lines 259-264 are Methods, not results, since they refer to the dates on which the visits to the different rivers have been made. The choice of the beginning of the sampling period should be justified. It starts in mid-June (June 9) in some reach, and in mid-July in others ??, but the start of emergence of the target species in the south of France is earlier, in May (early May for M. splendens, mid-May for for O. curtisii, late May for G graslinii, according to Boudot, J-P, & Kalkman (eds.) 2015 Atlas of the European dragonflies and damselflies. KNNV publishing, the Netherlands). The exuviae of individuals first emerged could have disappeared by the time the first visit was made. Mid-July is a start of sampling periods clearly inappropriate for these species. The global time of first, second and third visits overlaps: the first visits for some reachs overlaps with the second for others, and the second for some reachs overlaps in time with the third for others. This should be taken into account when performing the comparative analysis on the species located in the first, second and third visits in each reach.
Not all conclusions are supported by results/analyses of the manuscript. The authors do not discuss differences between two-person and one-person kayaks, so it cannot be concluded that one-person kayaks are preferable.
The time of year in which the visits have been made has not been discussed, and may affect the low numbers of exuviae detected.
The use of kayaks implies that the kayaks must be disinfected when changing from a river to its tributaries or when changing hydrographic basins, in order to avoid transmitting invasive species or diseases from one section to another, but this is not indicated in the manuscript. ?
The low number of exuviae, the lack of some usual species, and whether or not this sampling is adequate for these species should be commented on in the manuscript.
Author Response
Hello,
Please find the answers to all your comments on the attached Word file (Please see the attachment).
The new version of the manuscript is also available.
We remain at your disposal and thank you for your feedback.
Best regards,
Loan Arguel.

Reviewer 2 Report
[Line 184 to 187] Delete "A. mixta, C. boltonii, G. similimus, L. fulva, O. 184 albistylum, O. brunneum, S. sanguineum/meridionalis and T. annulata were also detected but 185 were very few (less than ten exuviae in total) and were thus discarded from further 186 analysis." It has already been mentioned before.
[Line 188] Add "with more than 10 exuviae" after "[...] detected".
[Lines 196, 200 and 203] Review the format of the text.
[Line 199] Two times on the Tarn, not three.
[Line 366]. Add a space after "third visit".
Author Response
Hello,
Please find the answers to all your comments in the attached Word file (Please see the attachment).
You will also find the new version of the manuscript.
We remain at your disposal and thank you for your comments.
Best regards,
Loan Arguel.

Reviewer 3 Report
The study addresses very important and interesting issues associated with the monitoring of insect populations, especially endangered species.
The study is well written, with a well-defined sampling and statistical design and in accordance with the proposed objectives.
I have some general considerations that I think are extremely important for the enrichment of the work.
I) The accurate introduction of information about the works that have already been done using exuvias. Already exist? where were they performed? What were the main goals. I suggest that a paragraph be added to the introduction mentioning this information.
II) Is it possible to add images of the three main species used in the study?
III) In our results the authors present data that show that there was a decrease in the number of species found from the first to the third visit. And the same happened for the abundance of the species found. However, in the discussion the authors do not return to these results. I suggest that this information be discussed at study.
Author Response
Hello,
Please find the answers to all your comments in the attached Word file.
You will also find the new version of the manuscript.
We remain at your disposal and thank you for your comments.
Best regards,
Loan Arguel.

Round 2
Reviewer 1 Report
The authors have commented and justified in the text of the manuscript the problems indicated for the previous version.
In the new version of the results maps (Figure 3 in this manuscript version) the authors put a similar colour to shown reachs with no-detection of exuvies. I find this presentation confusing; the previous version was clearer, with the points where exuviae had been sampled but the species had not been located marked with a white circle delimited by a black circumference.
In the Data Availability Statement, authors write "Not applicable". However, the manuscript is based on data from collected exuviae, which are not provided. The manuscript does provide the spatial distribution of the data for the three dragonfly target species of the Habitats Directive (Figure 3), but not the specific data of these records or the data of the other accompanying odonata species (which are used in the manuscript). It would be nice if the authors provided the dataset of these observations as suplementary file (a compatible DarwinCore file would be suitable), or deposit that dataset in some repository. It is not indicated whether the exuviae have been deposited in a public collection, or in a private collection of the authors, or whether the specimens have been destroyed and not preserved.
I think the rest of the questions have now been adequately presented and discussed, and the limitations (of time and weather) of the work have been made explicit. I think that the work can contribute to showing an adequate monitoring methodology for the dragonfly species G. graslinii and O. curtisii in rivers as wide as those sampled by the authors.
Author Response
Dear reviewer,
Figure 3 has been modified in the article according to your last comment.
Please find enclosed a table containing the data, as supplementary materials.
Thank you,
Best regards,
Loan Arguel.